# Micronutrient Deficiencies and Stunting Were Associated with Socioeconomic Status in Indonesian Children Aged 6–59 Months

**DOI:** 10.3390/nu13061802

**Published:** 2021-05-26

**Authors:** Fitrah Ernawati, Ahmad Syauqy, Aya Yuriestia Arifin, Moesijanti Y. E. Soekatri, Sandjaja Sandjaja

**Affiliations:** 1Center of Research and Development for Biomedical and Basic Technology of Health, National Institute of Health Research and Development, Jakarta 10560, Indonesia; fitrahernawati@yahoo.com (F.E.); ayarifin@gmail.com (A.Y.A.); 2Department of Nutrition Science, Faculty of Medicine, Diponegoro University, Jawa Tengah 50275, Indonesia; 3Center of Nutrition Research (CENURE), Diponegoro University, Jawa Tengah 50275, Indonesia; 4Nutrition Department, Health Polytechnic Ministry of Health of Jakarta II, Jakarta 12120, Indonesia; moesijanti@yahoo.com; 5Persatuan Ahli Gizi (PERSAGI), Jakarta 12320, Indonesia; san_gizi@yahoo.com

**Keywords:** micronutrient deficiency, stunting, socioeconomic status, malnutrition, Indonesian children

## Abstract

Micronutrient deficiencies and stunting are known as a significant problem in most developing countries, including Indonesia. The objective of this study was to analyze the association between micronutrient deficiencies and stunting with socioeconomic status (SES) among Indonesian children aged 6–59 months. This cross-sectional study was part of the South East Asian Nutrition Surveys (SEANUTS). A total of 1008 Indonesian children were included in the study. Anemia, iron deficiency, vitamin A deficiency, vitamin D deficiency, and stunting were identified in this study. Structured questionnaires were used to measure SES. Differences between micronutrient parameters and anthropometric indicators with the SES groups were tested using one-way ANOVA with post-hoc test after adjusted for age, area resident (rural and urban), and sex. The highest prevalence of anemia, stunting, and severe stunting were found to be most significant in the lowest SES group at 45.6%, 29.3%, and 54.5%, respectively. Children from the lowest SES group had significantly lower means of Hb, ferritin, retinol, and HAZ. Severely stunted children had a significantly lower mean of Hb concentration compared to stunted and normal height children. Micronutrient deficiencies, except vitamin D, and stunting, were associated with low SES among Indonesian children aged 6–59 months.

## 1. Introduction

Sustainable Development Goal-2 (SDG-2) aims to eradicate the global burden of malnutrition [1]. Malnutrition is one of the primary causes of mortality in children less than five years of age [2]. Decreasing malnutrition is a challenge for many countries, mainly developing countries [3]. In Indonesia, malnutrition remains a significant problem among children under five years old, especially micronutrient deficiencies and stunting [4,5,6]. Both micronutrient deficiencies and stunting can influence physical and cognitive development in children and increase the risk of infection [7].

A previous study in Indonesia indicated a high prevalence of anemia and vitamin D deficiency [7]. Almost 60% of Indonesian children under two years old were reported to be anemic [7,8], whereas the national prevalence of anemia among children two to five years of age was 16.6%. This figure was higher than Malaysia (6.6%) and Thailand (13.7%). Additionally, the prevalence of iron deficiency levels and vitamin D deficiency was 15% and 40%, respectively [7]. The vitamin D deficiency was quite high in Indonesia, but this prevalence lies between Malaysia (47.5%) and Thailand (36.7%). However, the prevalence of vitamin A deficiency in Indonesia (0.9%) was the lowest compared to Malaysia (4.4%) and Thailand (2.1%) [7]. On the other hand, the prevalence of stunting is also high in Indonesia [6,9]. Indonesian Basic Health Survey (Riskesdas) stated that the prevalence of stunting was almost 31% in 2018 [10]. Compared with the Association of Southeast Asian Nations (ASEAN) countries, the prevalence of stunting in Indonesia is much higher [9]. Overweight prevalence among subjects in Indonesia (4.4%) was the lowest compared to Malaysia (9.8%) and Thailand (7.5%). The same situation was also found in obesity, and the prevalence in Indonesia (3.5%) was also the lowest compared to Malaysia (10.5%) and Thailand (8.8%) [7].

Many factors are known to be involved in the etiology of micronutrient deficiency and stunting. Previous studies in Korea and China showed that low socioeconomic status (SES) was linked to micronutrient deficiency, including anemia, iron deficiency anemia (IDA), and vitamin D deficiency [11,12]. Moreover, some studies in Sri Lanka and Bangladesh found that low SES, overcrowding, and educated parents were associated with undernutrition among children [13,14,15]. In order to better address malnutrition in Indonesia, including micronutrient deficiencies and stunting, detailed information of the basic determinant factors is needed to design a more effective intervention/approach. Accordingly, there is a need for an in-depth understanding between micronutrient status and anthropometric indicators with SES in Indonesia. Therefore, the objective of this study was to analyze the association between micronutrient deficiencies (anemia, iron, vitamin A, and vitamin D) and stunting with socioeconomic status (SES) among Indonesian children aged 6–59 months.

## 2. Materials and Methods

### 2.1. Subjects and Study Design

The South East Asian Nutrition Survey (SEANUTS) was a multicenter study in nutrition funded by FrieslandCampina, The Netherlands. The SEANUTS was conducted in Indonesia, Malaysia, Thailand, and Vietnam in 2011. The SEANUTS in Indonesia utilized a cross-sectional study in 48 of 440 cities/districts in 2011 [7,9]. A multi- stage cluster sampling, stratified for area of residence (urban/rural), sex, and age was performed [7,9]. Details of the sampling procedure are described elsewhere [16]. A total of 1008 children aged 6–59 months living in rural (57.64%) and urban (42.36%) areas were included in the study. The participants were representatives of the target population. Given resource constraints in this study, blood samples for hemoglobin (Hb), serum ferritin, serum retinol, and serum 25-hydroxy vitamin D (25OHD) were taken in sub-samples of 1008, 475, 489, and 103 subjects, respectively, after being examined by medical doctors. Sub-samples were taken based on district in which each district consisted of two villages. From these villages, one was randomly selected for blood analysis. All samples aged 6 to 23 months and 24 to 59 months who lived in the area had their blood taken after examination by a local doctor to determine eligibility. Sub-samples were used due to budget limitation, and these sub-samples were chosen proportionately to represent age groups. Moreover, anthropometric measurements, including length and height, were taken for 983 children because some of them refused to be measured [7].

The study followed the guidelines of the Helsinki Declaration for human research. The design and methodologies were approved by the Committee of Health Research Ethics, the National Institute of Health Research and Development, the Ministry of Health of the Republic of Indonesia, number LB.03.02/KE/6430/2010, and the Ministry of Home Affairs, number 440.02/1751.D.I. The study was registered in the Netherlands Trial Registry (NTR 2462). Explanation of the study and the procedures applied, as well as the possible side effects and its management, was given to the parents before written informed consent was obtained.

### 2.2. Anthropometric Data

The length was measured supine using a flat wooden measuring board in children below two years of age. Height was measured using a wall-mounted stadiometer accurate to 0.1 cm in children aged two years old and older. All the measurements were done in duplicate with an accuracy of 0.1 cm. The average value was used in the calculations [9]. Height for age Z-scores (HAZ) was calculated using the WHO Anthro Software version 1.0.3 (https://www.who.int/tools/child-growth-standards/software, accessed on 25 June 2012) [17]. We used the WHO Child Growth Standards 2006 [18] to define severe stunting (HAZ < −3) and stunting (HAZ < −2).

### 2.3. Biochemical Indicators

Blood samples for Hb measurements were taken through the capillary blood procedure in children younger than two years old and from venipuncture in older children. Five milliliters of blood was taken from venous, then two milliliters of blood was inserted into an EDTA-coated tube, and the rest went into a tube without EDTA. EDTA blood was used for Hb, plain blood was centrifuged to take serum, and serum was used for ferritin, vitamin A, and vitamin D analysis. All the tube samples, especially serum retinol, were covered with aluminium foil, to protect from UVL. Hemoglobin concentration was measured with the HemoCue Hb 201 (HemoCue Diagnostics B.V. in all children). Anemia was defined as Hb concentrations of <110 g/L for children aged between 6–59 months. Serum ferritin was measured with Immunochemiluminescence ECLIA, Roche Cobas e 601; Roche Diagnostics. Iron deficiency was defined as serum ferritin concentrations of <12 µg/L for children under aged 6–59 months. Serum retinol was measured with an HPLC-UV detector, Agilent, 1200; Agilent Technologies (Santa Clara, CA, USA) (all-trans-retinol), and Serum 25(OH)D was measured with ELISA, IDS 25-Hydroxy Vitamin D; Immunodiagnostic Systems (D3 and D2 metabolites). Children with serum retinol concentrations of <0.70 µmol/L and circulating 25 hydroxyvitamin D <50 nmol/L were considered vitamin A or vitamin D deficient, respectively [7,19].

### 2.4. Socioeconomic Status

Structured questionnaires were employed to obtain information regarding income, education, housing type, flooring, ventilations, type of walls, ownership of valuable goods, and electronic appliances as well as type of household sanitation facilities. Socioeconomic status was calculated and categorized into five groups or quintiles, namely: lowest, low, middle, upper middle, and upper. We used national guidelines from the Central Bureau of Statistics (Indonesia) to categorize the SES [20]. The components of the SES included income, education, house (type, status, and valuable goods), and electricity [20]. Details for the data collection methodology and wealth classification were published earlier [16,21].

### 2.5. Data Analysis

Data were analyzed using SPSS version 24 (IBM Corp., Armonk, NY, USA) [7]. Weight factors were based on the 2010 Census data on the number of children in specific age groups [7]. Chi-square test was used to analyze the differences between characteristic variables across the groups of SES. Differences between micronutrient parameters and anthropometric indicators with the groups of SES were tested using one-way ANOVA with the Duncan post-hoc test after adjusting for age, area resident (rural and urban), and sex. Values are presented as mean and standard deviation for continuous variables or *n* (%) for categorical variables with *p* < 0.05 as significant.

## 3. Results

Overall, the mean age was 31.7 ± 16.1 months. The characteristics of the children in the five SES groups are presented in Table 1. Most of the children with lowest SES lived in rural areas (85.4%). The highest prevalence of anemia and iron deficiency was found in the lowest SES group with 45.6% and 16.4%, respectively. The highest prevalence of serum retinol deficiency was found in the upper middle SES group with 5.5%, but it does not increase gradually as SES decreases. Surprisingly, the highest prevalence of vitamin D deficiency was found in the middle SES group (60.9%). Moreover, the highest prevalence of stunting (HAZ < −2) and severe stunting (HAZ < −3) was found in the lowest SES group, with 29.3% and 54.5%, respectively.

The differences between micronutrient parameters and anthropometric indicators across the groups of SES are shown in Table 2. Children from the lowest SES group had significantly lower Hb (112.0 ± 13.2 g/dL), ferritin (30.9 ± 19.9 µg/L), retinol (1.28 ± 0.41 µmol/L), and HAZ (−1.77 ± 1.30). Differences in micronutrient status across HAZ indicator are shown in Table 3. Severely stunted children had significantly lower Hb concentration (110.8 ± 14.0 g/L) compared to stunted (114.0 ± 11.4 g/L) and normal height children (114.6 ± 13.2 g/L). In addition, children with normal height had significantly higher retinol concentration (1.54 ± 0.55 µmol/L) compared to severely stunted children (1.32 ± 0.39 µmol/L). However, ferritin and 25(OH)D concentrations were not significant in difference between normal height, stunted or severely stunted.

## 4. Discussion

This study provided insight into the relationship between micronutrient deficiencies and stunting regarding SES. The prevalence of anemia, iron deficiency, stunted, and severely stunted proved highest among the lowest socioeconomic groups 45.6%, 16.4%, 29.3%, 54.5%, respectively. However, the trend was not found in vitamin A deficiency prevalence. This may be due to the different sample sizes, particularly the smaller sub-samples for some micronutrient analyses. The results were similar to previous studies in middle and lower-income countries where anemia is more prevalent in children from lower SES groups [22]. A study in the Lancet by Balarajan et al. included nationally representative demographic and health surveys undertaken in 32 selected low-income and middle-income countries that conduct these surveys [23]. They showed that SES, especially household wealth, was significantly associated with anemia [23]. A previous study showed that children living in the lowest wealth quintile had significantly lower levels of hemoglobin [23], ferritin [24], retinol [25], and HAZ [9] than those in the highest wealth quintile [9,23]. They found that family income is considered an important determinant of micronutrient status [23] and anthropometric indicators [9]. When household income increased, the prevalence of anemia, IDA, and stunting decreased, and serum Hb and ferritin levels increased [11].

Iron, in the form of ferritin, is stored in the body. The body will take from these ferritin stores if the dietary iron needs are not reached. Chronic dietary iron insufficiency will deplete iron stores (mostly in the liver) as reflected by lower circulating ferritin. After iron depletion, hemoglobin synthesis is affected, lowering hemoglobin concentration progressively. This condition will cause iron-deficiency anemia. It can be prevented and treated by consuming foods high in iron, especially animal-based foods, because it contains iron in heme form, which is absorbed better than the non-heme iron form found in plant-derived foods [19,26]. However, animal-derived foods are often more expensive than plant-derived foods [26]. Therefore, the low consumption of animal-based foods as well as consumption of plant-based foods that also contain iron absorption inhibitors (e.g., phytate, oxalate, and polyphenols) may cause iron deficiency and anemia in the low SES group [26,27]. Previous studies have also found that high vitamin A deficiency within low SES groups, even though vitamin A also plays a role in the process of hematopoises and mobilization of iron in the body; thus vitamin A deficiency will aggravate iron status in the body [24].

These statements are supported by the results of the SEANUTS study in Indonesia regarding food consumption showing that animal protein and milk intake are positively correlated to SES [28]. The study also highlighted that only around 30% of households from the lowest SES had access to adequate sanitation facilities [28]. Moreover, lower SES is often related to poor living conditions, such as inadequate access to clean water and sanitation facilities, thus increasing the risk of infection and then increasing risk of developing anemia [29]. Interestingly, in this study, children in the lowest SES group had the highest iron deficiency but not significant.

The successful countrywide vitamin A supplementation program might offer an insight into this result. Serum retinol levels in children under five years old were higher in those who had received supplementation regularly than those who did not [29,30]. However, it is important to note that the lower education level among caregivers was the reason for missing doses in vitamin A supplementation. Hence, it might be useful for the future supplementation program [29,31]. It is also important to develop awareness about the health importance of vitamin A supplementation. Besides, the current cooking oil fortification program with vitamin A should offer an alternative solution. Fortification of oils with vitamin A is one of the low-cost and effective ways to improve vitamin A intake, reducing the risk of vitamin A deficiency in developing countries [32].

From this study, children from the middle SES families showed the lowest mean 25(OH)D concentration. However, no significant difference between vitamin D and SES groups was found. This may be due to the small sample size, particularly the smaller sub-samples for serum vitamin D analysis. Vitamin D is mostly made in the skin supported by sunlight exposure. Moreover, pigmentation of the skin is also responsible for vitamin D status [7,33,34]. Future studies regarding vitamin D should include coverage across the socioeconomic spectrum and couple a nutritional approach with sunlight exposures.

Regarding anthropometric indicators, a higher HAZ was correlated with a higher SAS. Moreover, this study revealed that stunted children had a higher risk of anemia than children with normal height, the same result was also found in other studies [35,36]. Ayoya et al. found that child’s age, HAZ score < −2 and mother’s anemia predicted the occurrence of childhood anemia in 6–59-months-old children in Haiti [37]. We also found significant differences in the risk of vitamin A deficiency between different anthropometric indicators. These findings were in line with other studies showing stunting was associated with vitamin A deficiency [38].

This study has a strength. To the best of our knowledge, this is the first study to discuss the relationship between micronutrient deficiencies (anemia, iron, vitamin A, and vitamin D) and nutritional status defined by anthropometric measurement with SES in Indonesia, especially children under five years old. Thus, this study will provide insights for better targeting when it comes to nutrition intervention. However, as a consequence of using sub-samples from a larger study, it has to be kept in mind that micronutrient status, in this study, was determined from a relatively small set of samples. Hence, the results should be interpreted carefully. We did not measure zinc in this study. Zinc is associated with chronic malnutrition and linear growth [39]. The current study is a cross-sectional study; consequently, it is unable to explain causal relationships. Another note is that the study did not include analysis of data on food intake and outdoor physical activity; thus, it is unable to explain the role of both behaviors concerning micronutrient status.

## 5. Conclusions

This study shows that micronutrient status and anthropometric indicators have an association with SES among Indonesian children aged 6–59 months. Anemia, iron deficiency, and stunting were associated with low SES. However, the trend was not found in vitamin A deficiency. While vitamin D status shows no association with SES. In addition, severely stunted children aged 6–59 months are significantly associated with anemia. The study suggests doing more comprehensive nutrition programs to improve the micronutrient status of children based on SES. Additional studies are needed to explore the association of micronutrient status and stunting with SES using a longitudinal study.

## Figures and Tables

**Table 1 nutrients-13-01802-t001:** Characteristics of children in the five socioeconomic groups ^1^.

Variables	SES					*p*-Value
	Lowest	Low	Middle	Upper Middle	Upper	
Age (years), *n* = 1008	31.5 ± 15.6	33.1 ± 16.5	31.5 ± 16.1	31.9 ± 16.1	32.4 ± 16.5	0.976
Sex, *n* = 1008						0.483
Boys	144 (50.3)	98 (49.7)	107 (53.5)	89 (55.6)	76 (46.3)	
Girls	142 (49.7)	99 (50.3)	93 (46.5)	71 (44.4)	89 (53.7)	
Area of residence, *n* = 1008						<0.001
Urban	42 (14.6)	84 (42.4)	89 (44.5)	90 (56.3)	122 (74.4)	
Rural	244 (85.4)	114 (57.6)	111 (55.5)	70 (43.8)	42 (25.6)	
Hemoglobin, *n* = 1008						<0.001
Normal	156 (54.4)	108 (54.8)	115 (57.5)	104 (65.0)	123 (75.0)	
Anemia	131 (45.6)	89 (45.2)	85 (42.5)	56 (35.0)	41 (25.0)	
Serum Ferritin, *n* = 475						0.087
Normal	112 (83.6)	78 (83.9)	82 (85.4)	67 (94.4)	75 (92.6)	
Deficiency	22 (16.4)	15 (16.1)	14 (14.6)	4 (5.6)	6 (7.4)	
Serum retinol, *n* = 489						0.128
Normal	136 (96.5)	89 (94.7)	97 (99.0)	69 (94.5)	83 (100)	
Deficiency	5 (3.5)	5 (5.3)	1 (1.0)	4 (5.5)	0 (0)	
Serum 25(OH)D, *n* = 103						0.164
Normal	22 (71.0)	11 (57.9)	9 (39.1)	11 (64.7)	6 (46.2)	
Deficiency	9 (29.0) 31(100)	8 (42.1) 19(100)	14 (60.9) 23 (100)	6 (35.3) 17 (100)	7 (53.8) 13 (100)	
HAZ, *n* = 983						<0.001
Normal height	151 (22.8)	124 (63.6)	132 (68.4)	116 (74.9)	140 (85.9)	
Stunted	81 (29.3)	50 (25.6)	48 (24.9)	32 (20.6)	14 (8.6)	
Severe stunted	45 (54.5)	21 (10.8)	13 (6.7)	7 (7.5)	9 (5.5)	

^1^ Values are presented as mean ± standard deviation for continuous variables and n (%) for categorical variables.

**Table 2 nutrients-13-01802-t002:** Micronutrient parameters and HAZ across the five socioeconomic groups ^1^.

Variables	SES					*p*-Value
	Lowest	Low	Middle	Upper Middle	Upper	
Hemoglobin, g/L	112.0 ± 13.2 ^a^	113.3 ± 13.0 ^a^	113.3 ± 12.7 ^a^	115.7 ± 12.7 ^b^	118.7 ± 11.9 ^b^	0.002
Serum Ferritin, µg/L	30.9 ± 19.9 ^a^	33.4 ± 21.2 ^a^	34.4 ± 20.7 ^a^	43.6 ± 24.4 ^b^	44.1 ± 10.9 ^b^	<0.001
Serum retinol, µmol/L	1.28 ± 0.41 ^a^	1.35 ± 0.44 ^a^	1.56 ± 0.45 ^b^	1.67 ± 0.47 ^b^	1.70 ± 0.74 ^b^	<0.001
Serum 25(OH)D, nmol/L	56.1 ± 10.7	52.0 ± 11.7	52.2 ± 16.4	55.3 ± 11.1	52.5 ± 20.8	0.721
HAZ	−1.77 ± 1.30 ^a^	−1.65 ± 1.13 ^a^	−1.47 ± 1.12 ^ab^	−1.03 ± 1.39 ^c^	−0.77 ± 1.46 ^c^	<0.001

HAZ: height for age Z-score, SES: socioeconomic status. ^1^ Values are presented as mean ± standard deviation for continuous variables and *n* (%) for categorical variables. Values corrected for sex, age, and area of residence. ^a–c^ Different superscripts indicate significant differences across SES groups.

**Table 3 nutrients-13-01802-t003:** Micronutrient status across linear growth categories ^1^.

Variables	HAZ			*p*-Value
	Normal Height	Stunted	Severely Stunted	
Hemoglobin, g/l	114.6 ± 13.2 ^a^	114.0 ± 11.4 ^a^	110.8 ± 14.0 ^b^	<0.001
Serum Ferritin, µg/l	37.7 ± 24.2	33.6 ± 22.5	34.3 ± 19.6	0.598
Serum retinol, µmol/L	1.54 ± 0.55 ^a^	1.37 ± 0.47 ^a,b^	1.32 ± 0.39 ^b^	0.012
Serum 25(OH)D, nmol/L	54.1 ± 14.7	51.9 ± 13.4	56.3 ± 9.8	0.722

HAZ: height for age Z-score. ^1^ Values are presented as mean ± standard deviation for continuous variables and n (%) for categorical variables. Values corrected for sex, age, and area of residence. ^a–c^ Different superscripts indicate significant differences across SES groups.

## Data Availability

The data presented in this study are available on request from the first author. The data are not publicly available according to description of confidentiality and data sharing procedures described in the study’s informed consent and assent documents.

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
