# Peer review of "Micronutrient Deficiencies and Stunting Were Associated with Socioeconomic Status in Indonesian Children Aged 6–59 Months"

_nutrients, 2021, doi:10.3390/nu13061802_

Round 1

Reviewer 1 Report

Main concerns

In the current study, Ernawati and coauthors evaluated the associations among socioeconomic status (SES) and the risk of micronutrient deficiency and stunting in infant and young children aged 6 to 59 months in Indonesia. Biochemical markers and anthropometric data were analyzed and collected from all or subgroups of participants. As the current work is part of a cross-section study of which other results have been published before, not all details of data collection methods including how SES were categorized were provided in the current study. In addition, some terms used in questionnaire for assessment of SES were uncommon (see in minor comments listed below); additional explanation/clarification should be provided. Moreover, I disagree with authors as to how prevalence of micronutrient deficiencies and stunting were calculated and presented in the table 1. In the current study, the prevalence was calculated and presented as the percentage of deficient or stunted participants from each SES group in all deficient or stunted participants. This is incorrect as the total number of participants are different in each of the SES groups. The prevalence should be calculated as the percentage of deficient or stunted participants and total participants within each of the SES groups. All data presented in the parenthesis in table 1 should be recalculated, and corresponding results and interpretation should be rewritten.  The results of serum 25(OH)D should be interpreted with cautions given the much smaller pool of participants (103) compared to that in other measurements.     

Minor comments

L99: The cut-off of serum ferritin for iron deficiency should be 12 µg/L? Please correct.

L103: The cut-off of circulating 25 hydroxyvitamin D should be ~ 50 nmol/L. Please correct. 

L105-111: Some of the terms listed in the questionnaire (e.g. type of walls, flooring, and  are not commonly used for evaluation of socioeconomic status in other studies. Please provide more details about how SES was assessed in the current study and justify the use of these terms in evaluation of social economic status.  

Table 1:

  • “%” should be removed from some values in parenthesis.   
  • I would suggest to report the percentage of deficient and normal individuals for the measured variables within each SES group rather than reporting the percentage of either deficient or normal individuals across SES groups.

Table 2:

  • Superscripts for ferritin, retinol and 25(OH)D are not correct; please edit.
  • Round up P-value to 3 decimal places (or per requirements of the journal) or use P < 0.001; Check P-values in all the other tables and make same change accordingly.
  •  

Reviewer 2 Report

I have attached a word version of the manuscript with annotated comments/questions and edits. The major comments follow:

  1. The description of methods is insufficient. It could use similar language and previously published papers as done by one of the co-authors (Dr. Soekatri) in the 2020 paper on determinants of stunting in children 0.5 to 12 years.
  2. The references are numbered in the text but not in the bibliography section, which makes checking the appropriateness of citations difficult. The paragraph (lines  ) in the discussion is very confusion regarding the sources being used to support the statements therein. More descriptors (e.g., country of study, age group, etc.) are needed to assess whether the authors are comparing apples with apples. Please correct this.
  3. The discussion of the role of vitamin A and VA supplements to explain iron (ferritin) and anemia results is confusing and even contradictory. 
  4. The study aims to explore the association of basic determinants of MN deficiencies (vit. D, Vit. A, iron) and stunting.  However, only socioeconomic status is included in the analysis. The association between SES and nutritional status is a well known fact.
  5. Discuss the impact of sample size (particularly the smaller subsamples for some of the micronutrient analyses) on inferences about the association of SES with the prevalence of some of these deficiencies. Include this discussion as part of weaknesses of the study. Note: for Vit. A deficiency prevalence, for instance, the expected progressive decline of deficiency (%) as SES improves is not apparent, particularly for some cells with very few observations. The trend is very clear as expected for stunting and anemia data points, which are higher than for other micronutrients.
  6. It is very important for readers to understand how the SES was calculated (its components: income, household items, electricity, etc.?). Please describe the standard reference (and method) used to create the SES categories.

Round 2

Reviewer 2 Report

Thank you for the extensive and conscientious review.

The word version with comments is attached for your consideration. A number of said comments were not visible to authors in the pdf document submitted previously with my review.
